Effects of sex and joint action on voluntary activation

Ema Ryoichi r-ema@ssu.ac.jp 1
Suzuki Momoka 2
Kawaguchi Emi 3
Saito Itaru 2
Akagi Ryota 2 4
1 School of Management, Shizuoka Sangyo University , Iwata , Japan
2 College of Systems Engineering and Science, Shibaura Institute of Technology , Saitama , Japan
3 Graduate School of Human Sciences, Waseda University , Tokorozawa , Japan
4 Graduate School of Engineering and Science, Shibaura Institute of Technology , Saitama , Japan
Boonstra Tjeerd
Electronic publication date: 2018 Nov 15
Publication date: 2018
Volume: 6
Electronic Location ID: e5968
Received 2018 Jun 1; Accepted 2018 Oct 19
Copyright: ©2018 Ema et al.
Copyright year: 2018
Copyright holder: Ema et al.
License: This is an open access article distributed under the terms of the Creative Commons Attribution License, which permits unrestricted use, distribution, reproduction and adaptation in any medium and for any purpose provided that it is properly attributed. For attribution, the original author(s), title, publication source (PeerJ) and either DOI or URL of the article must be cited.
License URL: https://creativecommons.org/licenses/by/4.0/

Keywords: Knee extension, Plantar flexion, Sex difference, Twitch interpolation technique

Funding: The Nakatomi Foundation and JSPS KAKENHI JP16K01671 JP16H05918 This work was supported in part by the grant from the Nakatomi Foundation and JSPS KAKENHI Grant Numbers JP16K01671/JP16H05918. There was no additional external funding received for this study. The funders had no role in study design, data collection and analysis, decision to publish, or preparation of the manuscript.

==============================
The current study tested the hypothesis that voluntary activation during maximal voluntary contraction (MVC) conditionally depends on sex and joint action. Twenty-eight healthy adults (14 of each sex) performed knee extensor MVC and plantar flexor MVC at extended and flexed knee positions. Voluntary activation during MVC was assessed using a twitch interpolation technique. The voluntary activation during plantar flexor MVC at the extended knee position was significantly lower (P = 0.020, 95% confidence interval 1.4 to 14.6, Cohen’s d for between-subject design = 0.94) in women (88.3% ± 10.0%) than in men (96.2% ± 6.6%). In contrast, no significant sex differences were shown in the voluntary activation during knee extensor MVC (93.7% ± 5.9% (women) vs. 95.0%  ± 3.9% (men)) and during plantar flexor MVC at the flexed knee position (90.4% ± 12.2% (women) vs. 96.8% ± 5.6% (men)). The voluntary activation during knee extensor MVC was significantly higher (P = 0.001, 95% confidence interval 2.1 to 8.8, Cohen’s d for within-subject design = 0.69) than that during plantar flexor MVC at the extended knee position in women, whereas the corresponding difference was not observed in men. The results revealed that the existence of sex difference in the voluntary activation during MVC depends on joint action and joint angle.

Introduction

A magnitude of muscle activation during maximal voluntary isometric contractions (MVCs) is the major determinant of generated muscular force. Voluntary activation (VA%), determined by the twitch interpolation technique (Shield & Zhou, 2004), is an index often used to represent the magnitude of muscle activation.

It has been shown that the magnitude of VA% depends on movements at joints (joint action). For example, male participants demonstrated a lower VA% during knee extensor MVC than during plantar flexor MVC (Behm et al., 2002). As a possible reason for the difference, Behm et al. (2002) proposed the difference in the muscle fiber type compositions of agonist muscle groups, i.e., a difficulty in full recruitment of motor units during knee extensor MVC due to the relatively higher proportion of type II fibers of the quadriceps femoris compared with the triceps surae. The proportion of type II fibers of the vastus lateralis, the largest muscle among the quadriceps femoris in both sexes (Ema et al., 2017), is lower in women than in men (Hunter, 2014). Although the association between VA% and muscle fiber type composition is unclear, if the proposal by Behm et al. (2002) is correct, women may show a higher magnitude of VA% during knee extensor MVC than men. In contrast, the soleus, which has the largest physiological cross-sectional area among the triceps surae (Fukunaga et al., 1992), comprises mainly type I fibers even in men (Johnson et al., 1973), suggesting that any possible sex difference of the soleus fiber type composition will be smaller compared with the corresponding difference of the vastus lateralis. Moreover, given that knee flexion reduces the neural and mechanical contribution of the gastrocnemius to plantar flexion strength (Wakahara et al., 2007), it can be assumed that the sex difference in VA% during plantar flexor MVC, if any, will be small, especially at a flexed knee position. A training program induced a similar extent of strength gains between sexes despite a smaller magnitude of muscle hypertrophy in women (Lemmer et al., 2000; Melnyk, Rogers & Hurley, 2009), which may indicate greater neural adaptation by the training in women than in men. Considering that the magnitude of VA% before training intervention (Gondin et al., 2005) and its training-induced change (Ema, Saito & Akagi, 2018) were related to strength gain, an investigation of the above notions should promote better understanding of the underpinning mechanisms of training-induced strength improvement in men and women.

To the best of our knowledge, no studies have investigated the effects of sex and joint action on VA% during MVC simultaneously. In the current study, we determined VA% during knee extensor MVC and plantar flexor MVC at extended and flexed knee positions in both sexes. We tested the hypothesis that VA% during MVC conditionally depends on sex and joint action.

Methods

Participants

A sample size estimation (G*Power 3.1.7, Kiel University, Germany) was performed to detect a within-between interaction for VA%. The expected effect size, α, power, and correlation among repeated measures were set at 0.25, 0.05, 0.80, and 0.5, respectively. The estimation showed that 28 participants are required. It was proposed that physical activity and the existence of practice for strength testing affect the magnitude of VA% (Hunter, Pereira & Keenan, 2016). Therefore, we recruited untrained healthy young adults, and all participants visited our laboratory in advance for familiarization and practice in performing MVCs with the experimental setting for the right leg. A total of 28 adults (14 of each sex) with no habitual resistance exercises, knee or ankle injuries participated in the study (Table 1). We confirmed no significant sex difference in the magnitude of habitual physical activity, assessed with the long version of the International Physical Activity Questionnaire (Craig et al., 2003). The strength testing for knee extension and plantar flexion was performed on different days in random order among the participants. This study was approved by the Ethics Committee of the Shibaura Institute of Technology (Acceptance number: 16-008). All participants were informed of potential risks and the study’s purpose, and they provided written informed consent before participation.

Table 1 Physical characteristics of participants.

		Men (n = 14)	Women (n = 14)	
Age	years	23  ±  4	22  ±  1	
Height	cm	169.7  ±  4.4	157.6  ±  4.1	
Body mass	kg	62.0  ±  6.2	51.5  ±  6.6	
Physical activity	MET min/wk	3,108  ±  2,177	2,836  ±  2,542	
Notes.

MET, metabolic equivalent. Data are shown as mean ± standard deviation.

Evoked twitch responses

To provide insights into the effects of muscle fiber type composition on VA%, we investigated the twitch contractile properties, because the properties have been reported to be associated with the composition (Harridge et al., 1996; Hamada et al., 2000). Participants sat (for knee extensions) or lay supine (for plantar flexions) on the bench of an isokinetic dynamometer (CON-TREX MJ, PHYSIOMED, Germany) while being secured at the pelvis and torso to the dynamometer with nonelastic straps (Fig. 1). The knee and hip joint angles were set at 90° and 80°, respectively, for knee extension (anatomical position = 0°). For plantar flexion, the knee joint angle was 0° (K0) or 90° (K90) and the ankle joint angle was 0° (Kennedy & Cresswell, 2001). The centers of rotation of the dynamometer and the right knee/ankle joints were visually adjusted. Using a constant current variable voltage stimulator (DS7A; Digitimer Ltd, Welwyn Garden City, UK), the quadriceps femoris and triceps surae twitch responses were obtained with rectangular pulses of 1 ms. For the quadriceps femoris, to percutaneously stimulate the femoral nerve, a cathode (2 × 2 cm) was placed in the femoral triangle, and an anode (4 × 5 cm) was placed in midway between the superior aspect of the greater trochanter and the inferior border of the iliac crest. For the triceps surae, the tibial nerve was stimulated percutaneously in the popliteal fossa with the cathode and over the ventral aspect of the thigh with the anode. The supramaximal stimulus intensity was determined by increasing the current intensity until plateaus in the twitch torque occurred. Thereafter, five supramaximal twitch responses at a higher current (≥20%) were obtained every 10 s. Torque signals were recorded at 4 kHz and stored in a personal computer after A/D conversion (PowerLab16/35; ADInstruments, Bella Vista, Australia). After low-pass filtering the signal at 500 Hz, contraction onset was manually identified as described previously (Ema, Saito & Akagi, 2018). A previous study used the time to peak twitch torque (TPT), i.e., the duration from torque onset to peak twitch torque, as an index of estimated muscle fiber type composition (Kubo & Ikebukuro, 2010). Moreover, TPT was associated with muscle fiber type composition (Hamada et al., 2000). However, TPT is possible to depend on the magnitude of the peak value of twitch torque, making it unsuitable for comparisonsbetween sexes and between different joint actions. Therefore, we determined the twitch torque at 50 ms from torque onset relative to the peak value of twitch torque (normalized Twitch0−50) (Balshaw et al., 2016), and used this metric as the index of estimated muscle fiber type composition. The data were averaged across five contractions.

Figure 1 Schematic illustration of the experimental setup for knee extension (A), plantar flexion at extended (B) and at flexed knee positions (C).

VA% evaluations

After several warm-up contractions involving two maximal MVCs, participants performed knee extensor/plantar flexor MVCs for 3 s two times. A one-minute rest was provided between contractions. Verbal encouragement was provided during the contractions. When peak torque was visually observed on screen and 2 s after the MVC, supramaximal triplet stimulations at 100 Hz were interpolated. If the difference in the peak value of torque before stimulation was above 10%, an additional contraction was requested. The VA% was calculated as follows: (1 − [superimposed triplet torque/potentiated resting triplet torque]) ×100 (Miyamoto et al., 2012). The mean of the two trials was used for subsequent analyses.

Statistical analyses

Statistical analyses were performed using SPSS version 22 (IBM, Armonk, NY, USA). All data are shown as means ±  standard deviation. The significance level was set at P <  0.05. A two-way analysis of variance (ANOVA) with one between-group factor (sex; men and women) and one within-group factor (joint action; knee extension, plantar flexion in K0 and K90) was conducted on dependent variables. When a significant interaction of sex × joint action was shown, follow-up ANOVAs with Bonferroni multiple-comparisons were used. To examine the magnitude of the difference in variables, Cohen’s d (between- or within-subject designs, (Lakens, 2013) was calculated as an index of effect size (ES), and 95% confidence interval of the difference were determined. The thresholds for interpretation of ES were 0.20, 0.60, and 1.20 for small, medium, and large (Hopkins et al., 2009). We considered the differences to be substantial if both ES >0.60 and P <0.05.

Results

Figure 2 shows VA% during MVC. There was a significant sex × joint-action interaction (F[2, 52] = 3.221, P = 0.048) and significant main effect of sex (F[1, 26] = 4.27, P = 0.049) but not of joint action (F[2, 52] = 1.198, P = 0.310). Regarding the sex difference, VA% during plantar flexor MVC in K0 was significantly higher in men (96.2% ± 6.6%) than in women (88.3% ±  10.0%), whereas no significant sex difference was found for VA% during knee extensor MVC (95.0% ± 3.9% [men] vs. 93.7% ±  5.9% [women]) or plantar flexor MVC in K90 (96.8% ± 5.6% [men] vs. 90.4% ± 12.2% [women]), with the observed effects as small to medium (Table 2). For joint action dependency, VA% during knee extensor MVC was significantly higher than that during plantar flexor MVC in K0 in women, with the effect being medium, but not in men. No corresponding differences were shown between other joint actions in each sex.

Figure 2 Scatterplots showing individual data of voluntary activation (VA%) during maximal voluntary contractions of knee extension, plantar flexion at extended (K0) and at flexed (K90) knee positions.

Solid line shows the mean value. *Indicates a significant difference between joint actions. † Shows a significant difference between sexes.

Table 2 Statistical results of the difference between sexes and between joint actions.

	P value	ES	95% confidence interval	
VA%				
Males vs. Females				
KE	0.501	0.26	−2.6 to 5.1	
PF in K0	0.020	0.94	1.4 to 14.6	
PF in K90	0.086	0.68	−1.0 to 13.8	
KE vs. PF in K0				
Males	0.421	0.24	−4.4 to 1.9	
Females	0.001	0.69	2.1 to 8.8	
KE vs. PF in K90				
Males	0.418	0.39	−4.6 to 0.9	
Females	0.152	0.36	−2.9 to 9.5	
PF in K0 vs. in K90				
Males	0.772	0.10	−1.9 to 0.7	
Females	0.299	0.19	−8.2 to 3.9	
Normalized twitch0−50				
Males vs. Females				
KE	0.545	0.25	−5.6 to 2.8	
PF in K0	0.545	0.08	−3.3 to 4.0	
PF in K90	0.545	0.33	−6.4 to 2.6	
KE vs. PF in K0				
Males	<0.001	2.00	8.3 to 15.1	
Females	<0.001	3.37	10.1 to 16.7	
KE vs. PF in K90				
Males	<0.001	1.87	8.0 to 16.8	
Females	<0.001	2.80	8.3 to 15.4	
PF in K0 vs. in K90				
Males	1.000	0.12	−1.4 to 2.7	
Females	1.000	0.35	−3.5 to 0.4	
Notes.

ES Cohen’s d for between- or within-subject designs

KE knee extension

Normalized twitch0−50 twitch torque at 50 ms from torque onset relative to the peak value of twitch torque

PF in K0 plantar flexion at extended knee position

PF in K90 plantar flexion at flexed knee position

VA% voluntary activation

The normalized Twitch 0−50 results are described in Fig. 3. A significant main effect of joint action (F[2, 52] = 90.321, P <  0.001) without a main effect of sex (F[1, 26] = 0.376, P = 0.545) or an interaction of the two factors (F[2, 52] = 0.607, P = 0.549, partial η2 = 0.023) was shown. The normalized Twitch 0−50 of knee extension was significantly greater than those of plantar flexions for both extended and flexed knee positions, with the effects being large.

Figure 3 Scatterplots showing individual data of twitch torque at 50 ms from torque onset relative to the peak value of twitch torque (Normalized twitch0−50) of knee extension, plantar flexion at extended (K0) and at flexed (K90) knee positions.

Solid line shows the mean value. *Indicates a significant difference between joint actions.

Discussion

The main finding of the current study was that the substantial sex difference in VA% was shown only during plantar flexor MVC at the extended knee position. Compared with men, women showed lower VA% during plantar flexor MVC at the extended knee position, and the magnitude of sex difference was interpreted as medium (ES = 0.94). In contrast, a clear sex difference in VA% was not observed during knee extensor MVC or during plantar flexor MVC at the flexed knee position. In addition, in women but not in men, VA% during plantar flexor MVC at the extended knee position was substantially different from that during knee extensor MVC. These results indicate that the sex difference in VA% during MVC depended on joint action.

The substantial sex difference in VA% was shown only during plantar flexor MVC in K0. The normalized Twitch0−50 was not significantly different between sexes for any joint action (Fig. 2), suggesting that the muscle fiber type composition is not a major factor for the observed sex and joint-action dependency in VA%. Co-contraction of the antagonist tibialis anterior during plantar flexor MVC might be the explainable factor for the current result. It was shown that compared with males, females exhibited higher magnitude of the tibialis anterior activation relative to medial gastrocnemius activation during the push-off phase of countermovement jumping (Márquez et al., 2017). During countermovement jumping, just before take-off, the fascicles of the medial gastrocnemius have been reported to contract quasi-isometrically (Kurokawa et al., 2003); therefore, the contraction type of the triceps surae may be partly similar between the current (i.e., isometric contraction) and previous (Márquez et al., 2017) studies. Because of reciprocal inhibition (Crone et al., 1987), co-contraction of the tibialis anterior during plantar flexor MVC could diminish the magnitude of triceps surae activation during plantar flexion. In contrast, the absence of a clear sex difference in VA% during knee extensor MVC (Fig. 2) is in line with previous studies (Krishnan & Williams, 2009; Lee et al., 2017), and Krishnan & Williams (2009) observed that the magnitude of hamstring activation as antagonists during knee extensor MVC was not different between sexes. Such notions may be also related to the substantially higher VA% during knee extensor MVC than during plantar flexor MVC in K0 only in women. Taken together, a possible sex difference in the antagonist activation may account for the current sex- and joint-action differences in VA%.

The lack of significant difference in VA% between joint actions in male participants is not consistent with the previous finding of a higher value of VA% during plantar flexor MVC than during knee extensor MVC in men (Behm et al., 2002). This discrepancy may be related to the difference in participant backgrounds. We recruited untrained participants because resistance training can affect the magnitude of VA% (Ema, Saito & Akagi, 2018). In contrast, Behm et al. (2002) examined subjects who participated in habitual resistance exercises or competitive sport activities. Although the kind of resistance training that the subjects had performed was not mentioned, it is possible that the effects of training on VA% during MVC differed between joint actions, likely resulting in the higher VA% during plantar flexor MVC than during knee extensor MVC (Behm et al., 2002).

Our findings may provide some implication regarding the training-induced strength gains. Previous studies demonstrated that greater muscle hypertrophy in men than in women after resistance training were accompanied by a lack of sex difference in knee extension strength gains (Lemmer et al., 2000; Melnyk, Rogers & Hurley, 2009). The previous results imply a greater neural adaptation in women than in men, because there was a negative correlation between VA% before training and the magnitude of strength improvement (Gondin et al., 2005). However, no significant sex difference in VA% during knee extensor MVC was found in the present study; therefore, it is difficult to explain the aforementioned results in terms of a sex difference in neural adaptations. In contrast, our data might suggest the greater training-induced increase in plantar flexion strength in women than in men, and the greater strength gains in plantar flexion than in knee extension in women; future attempts are required to clarify this subject.

Conclusion

The substantial sex difference in VA% was limited during plantar flexor MVC at the extended knee position, and only women showed joint action dependency in VA%. These results revealed that the existence of sex difference in VA% during MVC depends on joint action and joint angle.

Supplemental Information

Supplemental Information 1 Raw data

Click here for additional data file.

Additional Information and Declarations

Competing Interests

Author Contributions

Human Ethics

Data Availability

The authors declare there are no competing interests.

Ryoichi Ema conceived and designed the experiments, performed the experiments, analyzed the data, contributed reagents/materials/analysis tools, prepared figures and/or tables, authored or reviewed drafts of the paper, approved the final draft.

Momoka Suzuki, Emi Kawaguchi and Itaru Saito performed the experiments, analyzed the data, authored or reviewed drafts of the paper, approved the final draft.

Ryota Akagi conceived and designed the experiments, contributed reagents/materials/analysis tools, authored or reviewed drafts of the paper, approved the final draft.

The following information was supplied relating to ethical approvals (i.e., approving body and any reference numbers):

This study was approved by the Ethics Committee of the Shibaura Institute of Technology (Acceptance number: 16-008).

The following information was supplied regarding data availability:

The raw data is included in Figs. 2 and 3 and the Supplemental File.

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
