# Peer review of "Effects of sex and joint action on voluntary activation"

_PeerJ, doi:10.7717/peerj.5968_

## Round 0.1 · original submission · Minor Revisions

The manuscript has not gone to review yet, as we noticed that the statistical results are incompletely reported. All statistical results should be fully reported, including the test that was performed, the corresponding test statistic, degrees of freedom, the exact p-value (not p<0.05), and effect sizes. It is also recommended to overlay bar graphs with scatter plots showing individual data points, see e.g. https://doi.org/10.1371/journal.pbio.1002128.

Once you have included the requested information, we will send out the manuscript for peer review. Please be aware that the manuscript will still need to be evaluated by two external reviewers if you choose to resubmit.

---

## Round 0.2 · Major Revisions

Although the reviewers thought the manuscript is of interest, they have raised several questions that need to be addressed before the manuscript can be accepted for publication. In particular, the authors should expand the method section, clarify the statistical tests that were used, and consider the advise regarding the presentation and interpretation of effect sizes. They should also make sure that the conclusions clearly state the main findings of the study.

Reviewer 1 ·

Basic reporting

Please report additional references regarding the possibility that there are differences in motor unit firing rates between males and females. For example, Beck et al. (2005) reported that there were no differences in the patterns of response for global firing rates between males and females (assessed with mechanomyography). Also, that other potential differences in motor unit activation strategies were related more to absolute torque production than underlying mechanisms of neural control. Could the present study's findings be related to differences in torque production capabilities? For example, on average, men produces more torque than women.

Experimental design

Perhaps, to improve clarity and understanding of the stated aim (page 2, line 56), authors should define interaction. For example, instead of stating that an "interaction exists" the authors hypothesize that there is a "sex-dependent difference related to joint-action for VA%" or "VA% conditionally depends on both sex and joint-action". This suggestion is aimed to improve the clarity to the reader.

Concerns with the the 2 x 2 repeated measures ANOVA and its appropriateness for the given study. How was sex as a factor repeated? Would a mixed factorial ANOVA have been more appropriate?

Validity of the findings

Was the repeated measures ANOVA most appropriate model to examine the 2 (sex; women and men) x 2 (joint-action; knee extension and plantar flexion)? Are those both within factors? Or was this a mixed factorial ANOVA with sex being the between factor and joint-action being the within factor. I appreciate any and all details to help clear up my confusion. Specifically, my confusions sits with the sex factor and how that was repeated as a within factor.

How did the authors break down the significant sex x joint-action interaction (page 7, line 130)? Was it broken down or was the main effect of sex just examined?

If this sex difference truly exists, how will this knowledge improve training studies/training investigations?

As aforementioned, to what degree do the differences in VA% depend on the generalization that men are stronger than women?

Additional comments

The manuscript was well written, however, I recommend that results section is expanded. Specifically, how the authors broke down their significant interaction. In addition, the aim and scope (page 2, lines 51-57) be improved. Specifically, remove statistical terms such as 'interaction' and use a common definition to promote clarity. In addition, specific hypotheses such as "Based on previous findings (refs; refs; refs) men will have a greater VA% at this joint-action." This recommendation is suggested because that is what the actual repeated measures ANOVA is examining (differences in the means).

·

Basic reporting

'no comment'

Experimental design

'no comment'

Validity of the findings

'no comment'

Additional comments

ABSTRACT

Line 21 – I wonder what the added value of this sentence is. It should be clear that this is what happened based on the previously discussed findings.
23-24 – Be more precise with the conclusion. The results also revealed no sex differences in two of the tests. Do not solely focus on the positive effects.

INTRODUCTION

27 – A instead of THE
30 – Do you mean joint angle? Throughout the text the term joint action is used by isn’t defined anywhere.
37-38 – You assume a clear relationship between fiber type and %VA but provide no reference for it. This is quite a leap which requires a bare minimum of a few sentences showing that such a relationship does indeed exist.
40 – What does the following sentence “comprises mainly type I fibers even in men (Johnson et al., 1973) suggesting that any possible sex difference of the soleus fiber type composition will be small.” mean? The cited study provides no information about type 1 fiber distribution in females and you provide no citation showing that females and males have similar type 1 distribution.
Generally speaking, the set up for the question can be both clearer and stronger. First, a number of leaps were made without adequate citations. Mainly, the assumption about the relationship between fiber type and VA. Second, it is unclear why and how the proposed investigation “should promote better understanding of the sex dependency in training adaptation.” In what sense? You can considerably improve this part.

METHODS

Line 58 – Good job with calculating and reporting power analysis.
79 –Please add a picture of the set up for both sitting and lying positions.
109- How long was each contraction? How long of a break was provided between each one? When exactly were the twitches provides? 2 seconds into the MVC? When peak forces were visually observed on screen? Please provide exact details. Also, when exactly was the second twitch provided? Immediately post the MVC? Please clarify.
Overall, the methodology section is thin and should be expanded. Currently, if other scientists try to replicate your methodology it would be impossible for them as too many details are missing. Add more details so it would be easy for other to replicate.

STATISTICS

118 – So just to be clear, you conducted two different two-way ANOVA?
2 Gender x 2 muscle group
2 Gender x 2 joint angle
Please add Cohen’s d in addition to, or instead of partial η2, and provide the exact way it was calculated. Mainly, what the mean differences were divided by. This paper should be helpful.
https://www.frontiersin.org/articles/10.3389/fpsyg.2013.00863/full
127 – Please add an effect size measure to the differences such as Cohen’s d, % differences and also consider adding 95% confidence intervals. Avoid interpreting the results of your data solely based on p values. After you add effect size measures I recommend discussing the results in view of them as well. For example, “A small/medium/large magnitude, (non)statistically significant (ES = ; p =) main effect for condition was observed in which average VA% was X% higher in males (ABSOLUTE VALUE + SD) compared to the females (ABSOLUTE VALUE + SD).” This way of reporting is more informative and accounts for the magnitude of the effect.
Overall the results rely too heavily on p-values. I suggest moving the results into a table that includes p-values, 95% CI, Cohen’s d and absolute differences. It will be much easier to read and understand.

DISCUSSION

The first paragraph suggests that interpretation of the results was based solely on p-values. After redoing the stats section and including effect size measures, reinterpret the results in view of them as well and do so for the rest of the discussion as well.
163 – be careful not to give a reader the impression that results from other studies are the ones found in the present study. Make sure there is a clear distinction throughout the discussion as to when you are discussing results observed the present study and those reported elsewhere. Line 163, is an example of a confusing sentence that may give the wrong impression.
Overall I find the discussion section to be overly speculative, confusing, and indirect. It would be better if the authors systematically go over the findings one by one in a simpler and direct manner and in cases which the results were surprising to the authors I recommend stating that clearly. I also suggest avoiding overly speculative explanations. Finally, what is the added value of the results? How do they enrich the literature? Which gaps do the fill? What should be the next stage? What were the limitations?

CONCLUSION

The results, to my understanding, show no effect between genders other than with the knee position. This is not stated clearly in the conclusion.

---

## Round 0.3 · Minor Revisions

The reviewer raised some additional minor points that should be addressed before the manuscript can be accepted for publication.

·

Basic reporting

NA

Experimental design

NA

Validity of the findings

NA

Additional comments

The authors did a great job addressing the comments that I raised.

A few minor points:

Abstract

L.18 Consider adding some statistical information concerning the significant and non-significant effects.

L. 21. Same as above. Include some raw values and/or statistical information. Maybe distinguish between within/between comparisons for the ease of reading.

---

## Round 0.4 · accepted · Accept

The authors adequately addressed the remaining comments.